# Preliminary Experience with Oxygen-Enriched Oleic Matrix Breast-Shaped Dressings in Oncoplastic Breast Surgery

**DOI:** 10.3390/life16010051

**Published:** 2025-12-29

**Authors:** Agostino Rodda, Stefano Bottosso, Andrea Lisa, Nadia Renzi, Elisa Bascialla, Giulia Benedetta Sidoti, Germana Lissidini, Giovanni Papa

**Affiliations:** 1Department of Medical, Surgical and Health Sciences, Plastic and Reconstructive Surgery Unit, University of Trieste, 34149 Trieste, Italy; giuliabenedetta.sidoti@phd.units.it (G.B.S.); giovanni.papa@asugi.sanita.fvg.it (G.P.); 2Plastic Surgery Department, Ospedale di Cattinara, ASUGI (Azienda Sanitaria Universitaria Giuliano Isontina), 34149 Trieste, Italy; stefano.bottosso@asugi.sanita.fvg.it (S.B.); nadia.renzi@asugi.sanita.fvg.it (N.R.); 3Department of Biomedical Sciences, Humanitas University, Via Rita Levi Montalcini 4, Pieve Emanuele, 20090 Milan, Italy; andrealisamd@gmail.com; 4PhD Program in Applied Medical-Surgical Sciences, Department of Surgical Sciences, University of Rome “Tor Vergata”, Viale Oxford 81, 00133 Rome, Italy; 5Department of Plastic and Reconstructive Surgery, European Institute of Oncology, IRCCS, via Ripamonti 435, 20141 Milano, Italy; elisa.bascialla@ieo.it; 6Division of Plastic and Reconstructive Surgery, Department of Biotechnology and Life Sciences, University of Insubria, 21100 Varese, Italy; 7Division of Breast Surgery, European Institute of Oncology, IRCCS, via Ripamonti 435 Milano, 20141 Milan, Italy; germana.lissidini@ieo.it

**Keywords:** oncoplastic breast surgery, oxygen-enriched oleic matrix, advanced wound dressing, reactive oxygen species, wound healing, patient comfort

## Abstract

Wound-related issues such as delayed healing and patient discomfort remain common challenges in oncoplastic breast surgery and may negatively affect early postoperative recovery. This single-centre, retrospective, within-patient study explored the feasibility and safety of a breast-shaped polyurethane and polyester dressing impregnated with an oxygen-enriched oleic matrix, designed to provide a controlled, low-level release of reactive oxygen species involved in physiological tissue repair. Sixty patients undergoing unilateral lumpectomy with contralateral breast remodelling were included. The advanced dressing was applied to the oncologic breast, while standard premedicated patches were used on the contralateral side, allowing each patient to serve as her own control. Early postoperative outcomes, including wound dehiscence, infection, delayed healing, and qualitative user experience, were assessed descriptively over the first postoperative month. The oxygen-enriched oleic matrix dressing was well tolerated and associated with good skin hydration, comfort, and ease of use. No infections, hematomas, or reoperations were observed, and no relevant differences in early complication patterns emerged between the treated and control sides. Both patients and healthcare personnel reported favourable handling characteristics and comfort, with no device-related adverse events. These preliminary, hypothesis-generating findings suggest that oxygen-enriched oleic matrix breast-shaped dressings are a feasible and safe option for early postoperative wound management in oncoplastic breast surgery. Prospective, adequately powered multicentre studies are warranted to further investigate their potential role within standardized postoperative care pathways.

## 1. Introduction

Oncoplastic breast surgery allows for excellent reconstructive and aesthetic outcomes following lumpectomy. In a single surgical session, it enables remodelling of the excised breast, symmetrisation of the contralateral breast, correction of ptosis, and, in cases of mammary hypertrophy, a reduction in breast volume, thereby providing both functional and cosmetic benefits. This approach allows surgeons to maintain wide oncological excision margins, which translates into low rates of margin involvement [1]; moreover, contralateral breast reduction can reduce the risk of subsequent breast cancer [2].

The Wise-pattern incision technique allows for proper repositioning of the nipple–areola complex [3], but postoperative wounds at the junction of vertical and horizontal sutures are particularly vulnerable because of reduced blood supply. Additionally, the nipple–areola complex, depending on the chosen pedicle, can also be at risk. Several factors—including patient comorbidities, smoking, breast size, postoperative compliance, and dressing management—may contribute to wound-healing complications.

Postoperative wound complications in oncoplastic breast surgery, such as wound dehiscence, infection, and delayed healing, can significantly affect recovery. These events often require additional interventions [4], delay adjuvant therapies, reduce patient satisfaction, and increase healthcare costs. Dehiscence, with an estimated incidence of about 4%, together with other minor complications such as margin expansion (8%) [5], usually leads to unsatisfactory scars at 6–12 months and lower overall satisfaction.

Endogenous reactive oxygen species (ROS) play a physiological role in tissue repair, acting as secondary messengers that modulate immune-cell recruitment, angiogenesis, and fibroblast proliferation [6]. A balanced, low-level ROS production is essential for healing, whereas excessive oxidative stress may impair tissue repair.

Vegetable oils, particularly those rich in oleic acid, also contribute to wound recovery by restoring the lipid barrier and providing mild anti-inflammatory and antioxidant effects [7].

Dressings incorporating oxygen-enriched oleic matrices have previously been used in aesthetic and reconstructive breast surgery and in surgical wound care, demonstrating improved hydration, comfort, and ease of handling [8,9].

Building on these prior findings, the present study investigated the short-term performance of breast-shaped polyurethane and polyester dressings impregnated with an oxygen-enriched oleic matrix, designed to provide a controlled, low-level release of ROS that supports physiological tissue repair.

Conducted at Ospedale di Cattinara in Trieste, this retrospective, single-centre, within-patient study aimed to assess the feasibility and preliminary effectiveness of these devices in reducing minor complications after oncoplastic breast surgery, providing early clinical evidence for their potential role in postoperative wound management.

## 2. Materials and Methods

The study design was observational, retrospective, and single-centre, including 60 patients who underwent unilateral lumpectomy with contralateral breast remodelling between June 2024 and June 2025 at the Plastic and Reconstructive Surgery Unit, Ospedale di Cattinara (Trieste, Italy). The incision for sentinel lymph node biopsy or axillary lymph node dissection was always performed separately from the breast incisions. The polyurethane and polyester breast-shaped dressing impregnated with an oxygen-enriched oleic matrix was applied to the oncologic breast immediately after surgery, covering the entire breast and suture lines without the need for additional secondary dressings, while the contralateral breast received standard treatment with sterile premedicated patches (Figure 1). All patients underwent oncoplastic breast surgery with a Wise-pattern incision, and the device was applied regardless of comorbidities, to assess its general applicability across a representative clinical population. The devices evaluated are CE-marked in compliance with Regulation (EU) 2017/745; they were provided by the hospital and made available to all patients free of charge. These devices support wound healing by creating, through a controlled and gradual release of reactive oxygen species (ROS), local microenvironmental conditions that are favourable to tissue repair and unfavourable to bacterial proliferation. A postoperative bra was then applied to provide adequate compression. The first clinical control and dressing change were performed at 48 h, assessing each breast for signs of infection, seroma or hematoma, wound integrity, skin condition, and dressing adherence. If no complication occurred, the next control was scheduled on postoperative day 7, and the advanced dressing was then discontinued.

This design allowed each patient to serve as her own control, comparing the outcomes of the advanced dressing on the oncologic breast with those of standard treatment on the contralateral side. Because oncoplastic procedures aim for optimal symmetry between breasts, both sides share comparable tissue characteristics, incision design, and surgical conditions, thus providing a reliable within-patient comparison. Although each patient served as her own control, the study design was purely retrospective, and outcomes were analyzed descriptively. All patients were enrolled in the institutional 1-year follow-up programme to ensure long-term surveillance; however, the present analysis focused exclusively on early (within 30 days) postoperative outcomes, which are directly relevant to wound healing and dressing performance.

The inclusion and exclusion criteria are summarized in Table 1. Data on wound complications were collected at each follow-up visit. Specifically, the incidence of wound dehiscence (partial or complete separation of the incision), seroma (clinically detectable subcutaneous fluid collection requiring aspiration), infection (erythema, discharge, or warmth requiring antibiotics), delayed healing (persistence of un-epithelialized areas beyond 30 days), and the need for additional interventions were recorded. Data integrity was ensured through standardized collection methods by trained personnel.

Data were analyzed using descriptive statistics only, given the limited sample size and exploratory nature of the study. Continuous variables are reported as mean standard deviation, and categorical variables as absolute counts and percentages. No inferential statistical tests were performed, as the study was not powered for hypothesis testing. Patient and staff experiences with the device were also documented. The primary endpoint was the incidence of wound dehiscence and other early complications. Secondary endpoints included qualitative evaluation of the wound-healing process, patient comfort, and ease of use as reported by healthcare personnel. Although no validated questionnaires (e.g., BREAST-Q, VAS) were used, patients were asked about comfort aspects such as material softness, absence of adhesive tape, irritation or pruritus, and absorption capacity. Healthcare staff provided feedback on ease of application and removal, as well as adaptability during postoperative care. These qualitative observations, while non-standardized, provided clinically useful insights into patient satisfaction and the dressing’s practical usability.

## 3. Results

The study included 60 patients aged between 33 and 64 years (mean age, 47.3 years). The body mass index (BMI) ranged from 18.5 to 29.6, with a mean of 24.3. All oncologic-side surgeries utilized volume-displacement oncoplastic techniques, and no volume-replacement cases were included. All patients underwent contralateral breast surgery: 54 breast reductions (90%) and 6 mastopexies (10%). Fifty-four patients (90%) underwent axillary sentinel lymph node biopsy, while 6 patients (10%) underwent axillary lymph node dissection. The initial breast sizes ranged from A to F cups. Twelve patients (20%) were smokers, and three (5%) were diabetic; none of these patients experienced postoperative complications. No infections, seromas, hematomas, or reoperations occurred. Furthermore, no allergic reactions or skin pathologies were reported. Wound dehiscence occurred in six patients (10%), equally distributed between the oncologic and contralateral breasts. Delayed healing was also observed in six patients (10%), three in the oncologic breasts treated with the advanced dressing and three in the contralateral breasts treated with standard patches.

No cases of hematoma were recorded. Regarding fat necrosis, no cases led to delayed healing, dehiscence, or reoperation. However, late or deep fat necrosis was not assessed, as no postoperative imaging studies (e.g., ultrasound) were performed. Patient characteristics are summarized in Table 2.

The advanced dressing provided adequate wound hydration and optimal local conditions for healing but, as expected in this limited series, no apparent reduction in delayed healing or dehiscence rates was observed compared with standard treatment.

Nevertheless, the dressing demonstrated excellent comfort, adherence, and handling properties. Healthcare personnel reported that the device was easy to apply and remove, while patients appreciated its softness, non-adhesive design, and absence of irritation or pruritus.

Overall, the incidence and pattern of early complications were comparable between the oncologic and contralateral breasts, confirming the safety and feasibility of the oxygen-enriched oleic matrix breast-shaped dressing in the postoperative management of oncoplastic breast surgery.

## 4. Discussion

Oncoplastic breast surgery combines oncological and reconstructive principles to achieve both therapeutic and cosmetic outcomes, ensuring breast aesthetics while maintaining oncological safety. The remodelling of the excised breast, coupled with contralateral symmetrisation, offers significant functional and psychological benefits. Moreover, contralateral reduction mammoplasty can reduce the risk of future breast cancer and alleviate symptoms associated with macromastia. Traditional dressings, such as simple sterile patches, are commonly used after surgery to maintain wound cleanliness; however, they lack transparency for direct monitoring and offer no active contribution to the healing process. Advanced dressings such as hydrofiber–hydrocolloids [10] or negative-pressure wound therapies [11] can enhance moisture balance and antimicrobial protection, but at a considerably higher cost and with greater complexity of use. In this context, we evaluated a breast-shaped polyurethane and polyester dressing impregnated with an oxygen-enriched oleic matrix, designed to provide a controlled, low-level release of reactive oxygen species (ROS) that act as physiological mediators of wound healing, angiogenesis, and fibroblast activation. This controlled oxidative stimulus promotes a balanced microenvironment that supports tissue repair without inducing harmful oxidative stress. The oleic component contributes additional moisturizing and barrier-restoring properties, maintaining an optimal local environment for epithelialization.

The dressing was applied specifically to the oncologic breast, which is typically more prone to minor complications such as wound dehiscence and delayed healing [12]. In our series, no infections, seromas, or reoperations were observed, and the incidence of dehiscence and delayed healing was similar between the oncologic and contralateral breasts. Although the small sample size limits interpretation, these findings support the safety and feasibility of this approach in routine postoperative management. From a practical standpoint, the dressing was appreciated for its ease of application, full-breast coverage without adhesive tape, and transparency that facilitates early monitoring of skin flap and nipple–areola complex vitality. Surgeons valued the possibility of applying it directly in the operating room, immediately before the compression bra, thereby simplifying postoperative care. Healthcare staff also reported excellent handling, adaptability, and patient compliance, while patients highlighted comfort, absence of irritation, and ease of wearing the cup under a bra at home. The dressing provided excellent tissue hydration and moderate absorption of secretions, as shown in Figure 2, while maintaining an optimal local environment for healing. Although no validated tools (e.g., BREAST-Q, VAS) were used, these qualitative impressions provide useful preliminary insights into patient satisfaction.

The contralateral breast served as an internal control to reduce interpatient variability. This within-patient design offers a meaningful comparison, as both breasts share the same systemic and biological conditions and, in oncoplastic procedures, are reconstructed with the explicit goal of achieving symmetry. Consequently, the two sides exhibit similar incision patterns, tissue tension, and skin characteristics, making this approach suitable for exploratory evaluation. Nonetheless, minor differences in resection volume or vascularity may still introduce variability, which should be addressed in future studies through standardized surgical techniques or adjustment for confounders. The present results are consistent with recent global efforts to optimize outcomes and minimize complications in breast surgery, a topic that continues to attract international research interest across both reconstructive and aesthetic contexts [13]. Our findings contribute additional evidence to this field, suggesting that oxygen-enriched oleic matrix dressings may serve as a practical adjunct to enhance local healing conditions and patient comfort. A notable limitation of this study is the small sample size, which limits statistical power to detect differences in minor complications between treatment groups. However, the within-patient design helps mitigate interindividual variability and strengthens the internal consistency of the findings, even in a limited cohort. A hypothetical power analysis indicates that several hundred patients would be required to demonstrate statistically significant differences in minor complication rates. These estimates underscore the exploratory nature of the current study and the need for larger, multicentre collaborations to validate the present observations. Overall, this retrospective, single-centre experience demonstrates that the oxygen-enriched oleic matrix breast-shaped dressing is a safe, practical, and patient-friendly option for early postoperative care in oncoplastic breast surgery. While no reduction in complication rates was observed, the dressing’s handling properties, comfort, and potential benefits for wound monitoring justify further investigation in larger, adequately powered studies to confirm its potential clinical value.

## 5. Conclusions

In this single-centre retrospective experience, the polyurethane and polyester breast-shaped dressing impregnated with an oxygen-enriched oleic matrix was found to be safe, comfortable, and easy to use for postoperative wound care in oncoplastic breast surgery. The dressing was well tolerated and integrated smoothly into routine clinical practice, without an increase in early postoperative complications. Given the descriptive nature of the study, the limited sample size, and the retrospective design, these findings should be interpreted with caution and cannot be used to draw conclusions regarding clinical effectiveness or superiority over standard dressings. Nevertheless, the favourable handling characteristics and positive feedback from both patients and healthcare personnel support the feasibility of this approach in routine postoperative management. Further prospective, multicentre studies with larger cohorts, standardized clinical endpoints, validated patient-reported outcome measures, and longer follow-up are required to better define the role of oxygen-enriched oleic matrix dressings in postoperative wound care following oncoplastic breast surgery.

## Figures and Tables

**Figure 1 life-16-00051-f001:**
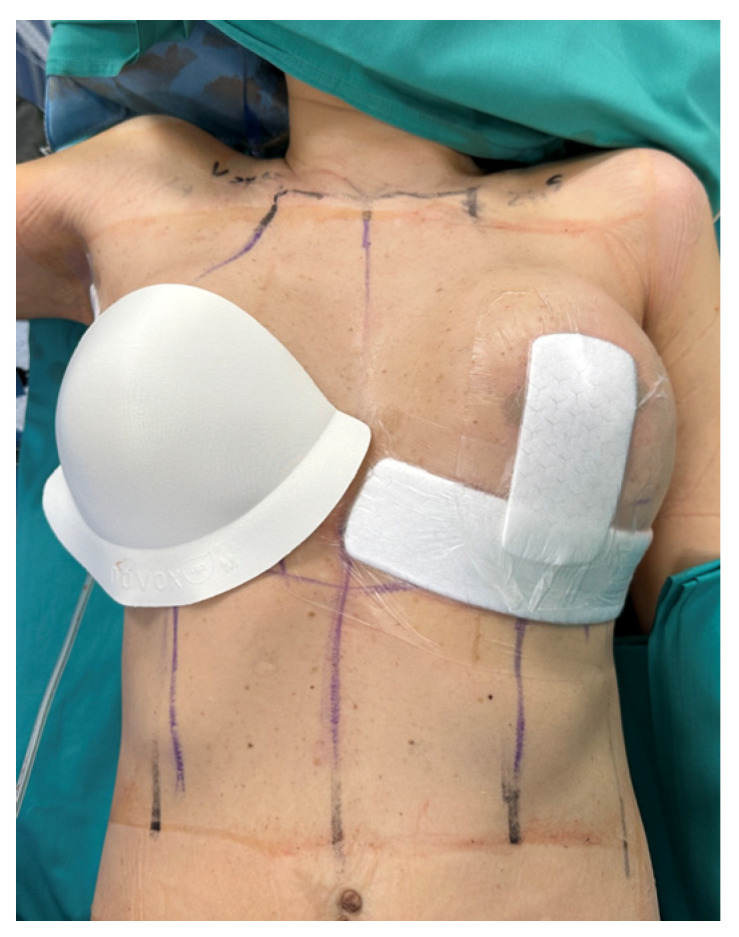
Immediate postoperative medication: the breast-shaped oxygen-enriched oleic matrix cup was applied on the right breast, while the contralateral side received standard dressings covering the suture lines.

**Figure 2 life-16-00051-f002:**
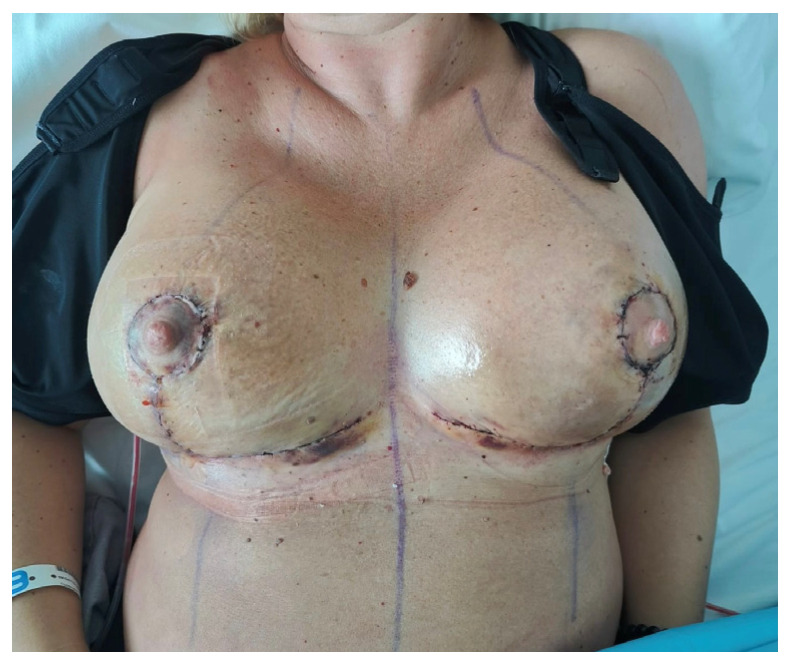
Postoperative assessment: the right breast was treated with standard dressings, while the left breast received the oxygen-enriched oleic matrix cup, showing visibly enhanced skin hydration and good tissue condition.

**Table 1 life-16-00051-t001:** Inclusion and exclusion criteria.

Inclusion Criteria	Exclusion Criteria
Unilateral lumpectomy	Known hypersensitivity to any device component
Contralateral breast remodelling Compliance to 1 year follow-up Ability to sign for informed consent	Any disorder compromising informed consent signing Previous breast radiotherapy

**Table 2 life-16-00051-t002:** Patient characteristics.

Parameter	Value
Number of patients	60
Age (years)	33–64 (mean: 47.3)
BMI	18.5–29.6 (mean: 24.3)
Breast Sizes (Cup)	A to F
Smokers	12 (20%)
Diabetes	3 (5%)
Allergies/Skin Pathologies	0 (0%)
Axillary sentinel lymph node biopsy	54 (90%)
Axillary lymph node dissections	6 (10%)
Infections	0 (0%)
Seromas	0 (0%)
Hematomas	0 (0%)
Need for additional surgeries	0 (0%)
Wound dehiscence	6 (10%)
Delayed healing	6 (10%)

## Data Availability

Data supporting reported results are available from the corresponding author upon reasonable request.

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
