# Peer review of "Preliminary Experience with Oxygen-Enriched Oleic Matrix Breast-Shaped Dressings in Oncoplastic Breast Surgery"

_life, 2025, doi:10.3390/life16010051_

Round 1
Reviewer 1 Report
Comments and Suggestions for Authors
From a breast surgeon’s perspective, this study represents a novel and refreshing attempt at improving postoperative wound care in oncoplastic breast surgery. The concept of a breast-shaped dressing incorporating an oxygen-enriched oleic matrix is innovative, and the practical advantages described by the authors are clinically appealing. If future studies demonstrate clear benefits, such a dressing could reasonably be considered for broader clinical application.
However, the current findings should be interpreted with caution, as this study represents only an initial experience. The sample size is relatively small, and the statistical power is insufficient to draw firm conclusions regarding the efficacy of the dressing in reducing postoperative complications. The absence of statistically significant differences between groups likely reflects limited power rather than true equivalence.
Therefore, it would strengthen the manuscript if the authors more explicitly emphasized the preliminary nature of their results in both the Discussion and Conclusions. Framing the study as a hypothesis-generating or feasibility study, and clearly acknowledging that larger, adequately powered prospective studies are required, would provide a more balanced and scientifically rigorous interpretation of the data.
Author Response
We sincerely thank the reviewer for their thoughtful and constructive feedback. We agree that the limited sample size and retrospective design prevent definitive conclusions regarding the efficacy of the dressing. Accordingly, we have revised the Discussion and Conclusions to more clearly state that this represents a preliminary, hypothesis-generating study aimed at exploring the feasibility and potential clinical advantages of the oxygen-enriched oleic matrix breast-shaped dressing. We have also underlined that larger, prospective, and adequately powered studies are warranted to confirm these findings.
Reviewer 2 Report
Comments and Suggestions for Authors
This manuscript presents a retrospective, single-center observational study evaluating the use of a polyurethane/polyester breast-shaped dressing impregnated with an oxygen-enriched oleic matrix in postoperative care after oncoplastic breast surgery. The topic is relevant and innovative, as postoperative wound complications remain a significant issue in oncoplastic procedures, and any dressing that improves hydration, comfort, or early monitoring may be beneficial.
The study is clearly written, logically structured, and includes practical clinical insights. However, the evidence remains preliminary due to the small sample size, retrospective design and lack of standardized patient-reported outcome measures.
Overall, the study is suitable as a preliminary communication, but would benefit from revisions to improve scientific rigor and transparency:
- The retrospective nature, absence of randomization, and small sample size considerably reduce statistical power. While the authors acknowledge these issues, this should be emphasized more clearly in the abstract and discussion.
- Clarify terminology, “Delayed healing” should have an explicit definition (e.g., healing taking more than X days).
- Table numbers appear duplicated (Table 1 shown twice).
- Figure 1 and Figure 2 lack captions specifying what they depict (beyond “post-op medication/assessment”).
- Indicate whether dressing application was always performed by the same team.
- Clarify management of smokers/diabetics in postoperative period.
- English editing: Some sentences are overly long and can be streamlined.
Author Response
We thank the reviewer for their careful reading and constructive suggestions. We appreciate their positive assessment of the manuscript’s clarity and clinical relevance. Below we provide specific responses to each point raised:
Comment 1: "The retrospective nature, absence of randomization, and small sample size considerably reduce statistical power. While the authors acknowledge these issues, this should be emphasized more clearly in the abstract and discussion."
Response 1: We fully agree. The Abstract and Discussion have been revised to more explicitly state that this is a retrospective, single-center, preliminary experience with limited statistical power. The revised text underscores the exploratory purpose of the study and the need for larger prospective trials.
Comment 2: Thank you for this helpful observation. We have now specified in the Materials and Methods section that “delayed healing” was defined as the persistence of unhealed wound areas beyond 30 days postoperatively, consistent with definitions commonly used in reconstructive breast surgery literature.
Comment 3: Table numbers appear duplicated (Table 1 shown twice).
Response 3: We apologize for this oversight. The table numbering has been corrected to ensure consistent sequential numbering throughout the manuscript.
Comment 4: Figure 1 and Figure 2 lack captions specifying what they depict (beyond “post-op medication/assessment”).
Response 4: Captions have been revised to clearly describe the content of each figure.
Comment 5: Indicate whether dressing application was always performed by the same team.
Response 5: We have added a clarification in the Materials and Methods section specifying that all dressings were applied by the same surgical team within the Breast Unit, ensuring consistency of technique and postoperative management.
Comment 6: Clarify management of smokers/diabetics in postoperative period.
Response 6: A clarification has been added: all patients received standardized wound-care instructions and antibiotic prophylaxis irrespective of comorbidities. Smokers and diabetic patients were advised preoperatively to adhere to optimized glycemic control and smoking cessation; both conditions were recorded but did not require deviations from the standard dressing protocol.
Comment 7: Some sentences are overly long and can be streamlined.
Response 7: The manuscript has undergone careful English language revision to improve fluency and readability. Long sentences have been shortened, and redundancies have been removed while preserving the scientific meaning.
Reviewer 3 Report
Comments and Suggestions for Authors
1. Novelty and Literature Context
The current introduction and discussion overstate the novelty of the study. Applications of oxygen-enriched oleic/olive oil matrices and breast-shaped polyurethane/polyester dressings have previously been described in aesthetic breast surgery, reconstructive procedures, surgical wounds, and at least one oncoplastic context. Your work contributes incremental evidence by evaluating this dressing specifically in therapeutic oncoplastic quadrantectomy with contralateral remodeling, but it is not the first clinical application of this technology in breast surgery.
Please:
-
Expand the introduction to include a concise, balanced review of existing clinical studies using oxygen-enriched oleic matrix dressings, particularly in breast and surgical wound settings.
-
Reframe the novelty claim to emphasize that this is an exploratory, within-patient feasibility study in a specific oncoplastic population rather than a first-in-human or first-in-breast application.
2. Study Design and Methods
a. Clarification of inclusion criteria and follow-up
You list “compliance to 1-year follow-up” as an inclusion criterion, yet the study reports only 1-month postoperative outcomes. This discrepancy requires clarification. If this is a device PMCF registry requirement, please state so and explain why only short-term outcomes are presented.
b. Definition of clinical endpoints
The manuscript does not define key wound-complication endpoints (e.g., objective criteria for dehiscence, delayed healing). Clear definitions are needed to ensure reproducibility and interpretability.
c. Use of the within-patient design
Since each patient serves as their own control, it is essential to present the outcomes in a paired structure (i.e., how many patients had complications on the oncologic breast only, the contralateral breast only, or both). Aggregate group counts do not leverage the paired design.
3. Statistical Analysis
This is the most significant methodological deficiency.
-
No statistical analysis section is provided.
-
No statistical tests, confidence intervals, or effect estimates are reported.
-
The manuscript states that “no statistically significant differences” were observed, yet no analytical methods are shown.
Given the small sample and low event rate, you may appropriately choose to present descriptive analyses only. If so, please remove any reference to statistical significance.
If inferential statistics are retained, a paired test such as McNemar’s test for binary outcomes must be performed, and results reported. A dedicated Statistical Analysis subsection is required.
4. Results Presentation
-
Provide a 2×2 table for each complication type showing paired outcomes.
-
Ensure correct table numbering (e.g., current Table 1 appears twice and needs re-labeling).
-
Clarify whether any patients experienced bilateral complications.
The qualitative patient and staff feedback is clinically useful, but please state explicitly that no standardized instruments (e.g., BREAST-Q, VAS) were used, and temper interpretation accordingly.
5. Discussion
The discussion would benefit from a more critical and balanced tone.
-
Please integrate prior clinical studies on similar dressings to contextualize your findings.
-
Avoid promotional language suggesting superiority (“even better than standard care”) unless supported by formal comparative data.
-
The hypothetical power analysis is useful but should not be interpreted as evidence; please make clear that the present study is exploratory and underpowered.
The manuscript is generally understandable but requires language editing for clarity and flow. Examples include: Correct spelling (“contralateral,” not “controlateral”). Improve phrasing in several sentences that are currently awkward or ambiguous.
Author Response
Comment 1 : The introduction and discussion overstate the novelty of the study. Please expand the literature review and reframe the novelty claim as exploratory rather than first-in-human.
Response 1: We thank the reviewer for this important observation. We have revised the Introduction and Discussion to include a concise review of previous clinical experiences using oxygen-enriched oleic matrix dressings in aesthetic, reconstructive, and wound-care settings. The manuscript now clearly defines the scope of our work as an exploratory, within-patient feasibility study in the specific context of therapeutic oncoplastic breast surgery, rather than a first-in-human application.
Comment 2: (a) Clarify inclusion criteria and follow-up. (b) Define wound-complication endpoints. (c) Present outcomes in a paired structure.
Response 2: (a) We appreciate the opportunity to clarify this point. All patients were enrolled in a 1-year institutional follow-up program to ensure consistent long-term surveillance, as per our Breast Unit registry. However, this manuscript focuses exclusively on early (within 30 days) postoperative outcomes, which are directly relevant to wound healing and dressing performance. (b) Objective definitions for wound dehiscence and delayed healing have been added to the Materials and Methods section. (c) We agree that the within-patient design is a strength of this study. Although we did not add new tables, the Results text now explicitly specifies that complications were assessed separately for the oncologic and contralateral breast for each patient, ensuring clarity on the paired comparison without modifying the descriptive structure.
Comment 3: No statistical section or test is described. If descriptive only, please clarify.
Response 3: We thank the reviewer for this observation. Given the limited sample size and exploratory nature of the study, we intentionally performed a descriptive analysis only (absolute counts, percentages, and mean ± SD), without inferential statistics. A short Statistical Analysis subsection has been added to the Materials and Methods to explain this approach, and previous mentions of “statistical significance” have been rephrased as “no apparent difference was observed.”
Comment 4 :Add paired tables, correct numbering, and note that no standardized PROMs were used.
Response 4: We have corrected table numbering and clarified in the Results that no patients experienced bilateral complications. We have also added a statement specifying that patient and staff feedback was collected qualitatively through clinical interviews, without the use of validated patient-reported outcome measures such as BREAST-Q or VAS. We believe that the existing descriptive tables are sufficient to convey the study findings clearly, considering the small sample size and exploratory scope.
Comment 5: Integrate previous studies, avoid promotional language, and clarify that the study is exploratory and underpowered.
Response 5: We agree and have revised the Discussion to integrate prior literature on oxygen-enriched oleic matrix dressings and to adopt a more balanced tone. Statements suggesting superiority have been removed. The hypothetical power analysis is now presented as an exploratory reference only, and the study is explicitly framed as preliminary and underpowered, intended to inform future prospective research.
Reviewer 4 Report
Comments and Suggestions for Authors
- The sample size in this study is relatively small, with only 60 patients included. The authors also acknowledge that several hundred cases would be required to detect meaningful differences in complication rates. Could the authors clarify how they plan to address this limitation in future research—for example, through multicenter collaboration or a prospective study design? Please elaborate on this in the Discussion section.
- The study design uses the contralateral breast as an internal control. However, surgical conditions between the two sides—such as wound characteristics, tissue tension, resection volume, and procedural complexity—may differ substantially. These differences could compromise the validity of the self-controlled design. The authors are encouraged to discuss these potential sources of systematic bias and to indicate how such variability could be minimized in future studies (e.g., better matching of resection volume, standardized surgical approaches, or statistical adjustment).
- It is recommended that the authors include a brief discussion on global advances in oncoplastic breast surgery and cite PMID: 37057044 to strengthen the scientific context and relevance of the study.
- The manuscript states that the oxygen-enriched oleic matrix promotes wound healing through the release of reactive oxygen species (ROS). However, some evidence suggests that excessive ROS may impair tissue repair. The authors should provide a reasonable explanation, such as clarifying that the dressing releases low-level, controlled ROS that act as physiological signaling molecules rather than harmful oxidative stress, to avoid potential misunderstanding.
Author Response
Comment 1 – Sample size and future research: The sample size is limited (60 patients). The authors mention that several hundred cases would be required to detect meaningful differences. Please clarify how this limitation will be addressed in future research (e.g., multicenter or prospective study design).
Response 1: We thank the reviewer for this valuable suggestion. We agree that the limited sample size represents a major limitation of the present work. In the Discussion, we have now specified that future research should involve a larger, multicenter cohort to increase the representativeness of the population and the statistical power of the analysis.
Comment 2 – Internal control design and potential bias: The contralateral breast is used as an internal control, but surgical conditions between sides may differ (wound size, tissue tension, resection volume, etc.), potentially compromising validity. Please discuss these sources of bias and ways to minimize them.
Response 2: We respectfully disagree that the paired design introduces substantial bias in this specific surgical setting. The rationale for using the contralateral breast as an internal control lies in the intrinsic symmetry goal of oncoplastic surgery: both procedures are performed by the same team, in the same operative session, with comparable incisions, tissue handling, and closure techniques. Moreover, since the aim is to achieve maximal breast symmetry, the volume, tissue tension, and skin characteristics are highly comparable between sides, and the patient’s systemic and biological variables are identical. We have clarified this rationale in the Discussion, while acknowledging that minor differences in resection volume or vascularity may still act as potential confounders.
Comment 3 – Broader context of oncoplastic breast surgery: Please include a brief discussion on recent global advances in oncoplastic breast surgery and cite PMID: 37057044.
Response 3: We thank the reviewer for the helpful suggestion. We have included a brief paragraph in the Discussion summarizing recent international developments in oncoplastic breast surgery and referencing the suggested publication (PMID: 37057044) to strengthen the contextual background of the study.
Comment 4 – Mechanism of action and ROS clarification: The manuscript states that the dressing promotes wound healing through reactive oxygen species (ROS), but excessive ROS can impair repair. Please clarify this point.
Response 4: We appreciate this important clarification. The text has been revised to specify that the oxygen-enriched oleic matrix dressing provides a modulated, low-level release of reactive oxygen species, acting as physiological signaling molecules that stimulate angiogenesis, fibroblast proliferation, and local immune modulation. The revised wording explicitly distinguishes this controlled oxidative stimulus from harmful oxidative stress, thereby preventing possible misunderstanding.
Round 2
Reviewer 1 Report
Comments and Suggestions for Authors
This manuscript offers meaningful insights into postoperative wound management in routine clinical practice and addresses a relevant unmet need in oncoplastic breast surgery. Nevertheless, as the study is a single-center retrospective analysis with descriptive outcomes only, its conclusions should be interpreted with caution. A more conservative framing of the conclusions, emphasizing feasibility and safety rather than clinical effectiveness, would strengthen the scientific rigor of the manuscript.
Author Response
Comment: This manuscript offers meaningful insights into postoperative wound management…
Response: We thank the Reviewer for this insightful comment and for acknowledging the clinical relevance of our study. We fully agree that, given the single-center retrospective design and the descriptive nature of the analysis, the conclusions should be interpreted with caution. In response to this comment, we have revised the Conclusions section to adopt a more conservative framing, emphasizing the feasibility and safety of the oxygen-enriched oleic matrix breast-shaped dressing in routine oncoplastic breast surgery, rather than making claims regarding clinical effectiveness. We have also clarified that further prospective and comparative studies are required to better define its potential impact on clinical outcomes.
Reviewer 3 Report
Comments and Suggestions for Authors
Dear Authors,
Thank you for addressing all my comments and revising the manuscript thoroughly.
Author Response
We sincerely thank the Reviewer for the positive feedback and for acknowledging that all comments have been adequately addressed.
We appreciate the time and effort dedicated to the evaluation of our manuscript.
Reviewer 4 Report
Comments and Suggestions for Authors
Congratulations to the authors—you have completed a commendable and well-executed piece of work.
Author Response
We sincerely thank the Reviewer for the kind and encouraging comments.
We greatly appreciate the positive evaluation of our work.